# Properties of Na_0.5_Bi_0.5_TiO_3_ Ceramics Modified with Fe and Mn

**DOI:** 10.3390/ma15186204

**Published:** 2022-09-07

**Authors:** Jan Suchanicz, Marcin Wąs, Michalina Nowakowska-Malczyk, Dorota Sitko, Kamila Kluczewska-Chmielarz, Krzysztof Konieczny, Grzegorz Jagło, Piotr Czaja, Bartosz Handke, Zofia Kucia, Patryk Zając, Klaudia Łyszczarz

**Affiliations:** 1Department of Mechanical Engineering and Agrophysics, University of Agriculture in Krakow, Balicka 120, 31-120 Krakow, Poland; 2Department of Bioprocess Engineering, Power Engineering and Automation, University of Agriculture in Krakow, Balicka 120, 31-120 Krakow, Poland; 3Institute of Physics, Pedagogical University, Podchorazych 2, 30-084 Krakow, Poland; 4Institute of Technology, Pedagogical University, Podchorazych 2, 30-084 Krakow, Poland; 5Faculty of Materials Science and Ceramics, AGH-University of Science and Technology, al.Mickiewicza 30, 30-059 Krakow, Poland

**Keywords:** Fe- and Mn-doped NBT, structural properties, dielectric properties, thermal properties, unpoled and poled states

## Abstract

Na_0.5_Bi_0.5_TiO_3_ (NBT) and Fe- and Mn-modified NBT (0.5 and 1 mol%) ceramics were synthesized by the solid-state reaction method. The crystal structure, dielectric and thermal properties of these ceramics were measured in both unpoled and poled states. Neither the addition of iron/manganese to NBT nor poling changed the average crystal structure of the material; however, changes were observed in the short-range scale. The changes in shapes of the Bragg peaks and in their 2Θ-position and changes in the Raman spectra indicated a temperature-driven structural evolution similar to that in pure NBT. It was found that both substitutions led to a decrease in the depolarization temperature T_d_ and an increase in the piezoelectric coefficient d_33_. In addition, applying an electric field reactivated and extended the ferroelectric state to higher temperatures (T_d_ increased). These effects could be the result of: crystal structure disturbance; changes in the density of defects; the appearance of (Fe_Ti_ˈ-), (Mn′_Ti_-V^••^_O_) and (Mn″_Tii_-V^••^_O_ )—microdipoles; improved domain reorientation conditions and instability of the local polarization state due to the introduction of Fe and Mn into the NBT; reinforced polarization/domain ordering; and partial transformation of the rhombohedral regions into tetragonal ones by the electric field, which supports a long-range ferroelectric state. The possible occupancy of A- and/or B-sites by Fe and Mn ions is discussed based on ionic radius/valence/electronegativity principles. The doping of Fe/Mn and E-poling offers an effective way to modify the properties of NBT.

## 1. Introduction

Due to their superior electromechanical properties, most ferroelectric materials used in devices covering a wide area of human activity are lead-based compounds [1,2,3]. Nevertheless, due to the need for more environmentally friendly materials, the development of lead-free alternatives has been in progress for the last few years.

Na_0.5_Bi_0.5_TiO_3_ (NBT) is seen as a potential replacement for lead-containing piezoelectric materials [4,5,6,7,8,9]. Although NBT reaches a maximum of electric permittivity at relatively high temperature T_m_ ≈ 320 °C, the depolarization temperature is lower (T_d_ ≈ 190 °C). Creating solid solutions with other perovskites or doping with various ions can affect both the depolarization temperature and piezoelectric response (d_33_), which is important from the point of view of potential applications. Unfortunately, usually, if T_d_ increases, d_33_ decreases.

Many properties of pure NBT such as its structural (both average and ion local scale), dielectric, thermal, pyroelectric, piezoelectric and optic properties have been studied, e.g., in [10,11,12,13,14,15]. These studies prove, among others, that a correlation between the presence of short-range structural disorder and the global structure, dielectric and ferroelectric properties of NBT exists.

Three dielectric anomaly peaks at T_d_, T_s_ and T_m_ are observed in NBT [16]. Here, T_d_ is the depolarization temperature, i.e., the temperature of the steepest decrease in remanent polarization, denoting a ferroelectric–relaxor transition, since above T_d_, the ferroelectric long-range order is no longer maintained. T_s_ denotes a “shoulder”-like low temperature dielectric anomaly, above which the polar nanoregions are subjected to large fluctuations making the formation of a ferroelectric phase difficult, even under high electric fields. At temperature T_m_, the dielectric permittivity reaches the maximum value. T_m_ is attributed to a diffuse phase transition of lower symmetry polar nanoregions (R3c) into higher symmetry (P4bm) ones and their thermal evolution.

With increasing temperature, NBT undergoes a transition from the rhombohedral to the tetragonal phase at about 260 °C and then to the cubic phase at 520–540 °C. In the temperature interval ~200–350 °C rhombohedral and tetragonal phases coexist. Neutron scattering, dielectric and pyroelectric measurements distinguish the temperature at 280 °C, below which unstable polar regions become stable [17,18]. They act as centers for the nucleation of the low-temperature ferroelectric phase, which occurs below about 200 °C. Transmission electron microscope studies have suggested that a rhombohedral–tetragonal phase transition may proceed through the intermediate modulated orthorhombic (Pnma) phase [7].

A few papers have reported the properties of Fe/Mn-doped NBT [19,19,20,21,22,23,24,25,26,27,28,29], however, most of them are limited to room temperature. There are no studies on the properties of these materials in a poled state. In the present paper, the effect of the addition of Fe and Mn to NBT ceramics on the temperature evolution of their crystal structure and dielectric and thermal properties in unpoled and poled states were systematically examined, analyzed and correlated.

## 2. Materials and Methods

NBT + xFe_2_O_3_ and NBT + xMnO_2_; x = 0, 0.05 and 0.1 (NBTFe05, NBTFe1 and NBTMn05, NBTMn1, respectively) ceramics were prepared via solid-state processing of high-purity amounts of Na_2_CO_3_, Bi_2_O_3_, TiO_2_, Fe_2_O_3_ and MnCO_3_ in the way described in [5]. The formation and quality of the samples was checked by X-ray diffraction (XRD) analysis.

XRD measurements were performed using Philips X’Pert Pro MD diffractometer (Amsterdam, Netherlands). Standard Bragg–Brentano geometry was applied with K_α1,2_ radiation (K_b_ line suppressed by Ni filter) from a Cu anode. All measurements were performed in air with a 0.017° step size in the 20–80° scanning range. Structural as well as qualitative and quantitative phase analyses were made by Phillips X’Pert High Score Plus version 3.0.5 software with implemented full-pattern fit by means of the Rietveld method. This Rietveld implementation was based on the source code of the program DBW3.2 from Wiles & Young [30].

The relative density of the specimens was measured by the Archimedes method and compared to the theoretical prediction. The obtained density was greater than 95% of the theoretical density. The microstructure analysis was undertaken by a scanning microscope (Hitachi S4700, Tokyo, Japan) with field emission and a Noran Vantage EDS system. The elemental distribution and composition were identified by elemental mapping and energy dispersive spectroscopy (EDS) [13].

The differential scanning calorimetry studies were carried out using a Netzsch DSC F3 Maia scanning calorimeter (Selb, Germany) in the temperature range from room temperature to 400 °C under an argon atmosphere at a flow rate of 30 mL/min. The specimen consisting of a single plate-like ceramic sample of mass 20 mg was placed in an alumina crucible. Measurements were performed upon heating and cooling at a constant rate of 10 °C/min.

Raman spectra of the powders were recorded using a Bio-Rad FTS 6000 spectrometer (Hercules, CA, USA) with an Nd-Yag laser system, where the 532 nm line was used as the excitation beam. The laser power was 200 mW. The spectra were collected with a resolution of 4 cm^−1^.

The dielectric measurements were carried out for silver electrode samples using a GW 821 LCR meter, Good Will Instruments Co., Ltd., New Taipei City, Taiwan in the temperature/frequency range from room temperature to 600 °C and from 100 Hz to 1 MHz, respectively. A measuring electric field of strength of 20 V cm^−1^ was applied. The data were collected regularly with a step of 0.1 °C upon heating and cooling at a rate of 1.5 °C/min using an automatic temperature controller.

The DC conductivity was measured using a Keithley 6517A electrometer (Cleveland, OH, USA) applying voltage within the range of applicability of Ohm’s law.

## 3. Results and Discussion

The SEM micrographs of the polished and chemically etched samples are shown in Figure 1.

As can be seen, the samples showed a dense microstructure and clearly visible grain boundaries. For the least-substituted samples, the grains tended to form cuboidal shapes and two sizes could be noticed: larger grains (~3 μm) of well-developed shapes with slightly rounded edges, and smaller ones (~1.5 μm) with more irregular and rounded shapes with blunt edges. For the higher-doped samples, no significant difference could be noticed in terms of the grain shape. The average grain size, which was determined by counting the number of grains across the diagonal, decreased after the Fe and Mn substitution into the NBT. The chemical diagrams of the constituent elements are shown in the inset of Figure 1. The backscattered electron (BSE) images and their corresponding elemental mappings show a relative homogeneous distribution of the elements in the samples. The results show also that the elemental proportions were in accordance with the experimental requirements.

The room temperature X-ray diffraction patterns consisted of a set of lines which were characteristic of a perovskite-type structure with rhombohedral symmetry (Figure 2).

The NBT was phase pure, whereas additional reflections were observed for both the Fe- and Mn-doped NBT. Bi_5_Ti_3_FeO_15_/Bi_5_Ti_3_MnO_15_ Aurvillius phase was identified as a secondary phase for both the F-e and Mn-doped NBT (4.7 and 0.8% for 0.5 and 1 mol% Fe-doped NBT, respectively, and 0.9 and 0.5% for 0.5 and 1 mol% Mn-doped NBT, respectively, see Table 1). The impurity peaks might have originated from the kinetics of the growth process [20]. In general, the R (weighted profile) values were lower than 10%, which indicated that the refinement was credible. The diffraction peaks initially shifted to lower 2Θ angles for 0.5%mol Fe-doped NBT, indicating a lattice expansion, and then shifted to higher angles for 1%mol Fe-doped NBT, indicative of a lattice contraction. The change in lattice parameters can be mainly due to the difference in ionic size of the dopant substituent in comparison to the host ion and their multiple oxidation states, which lead to the different bond length of Fe-O compared to the Ti-O and Na/Bi-O bonds and their changed force constants. The complex character of the (200)_c_ peak for the pure NBT indicated the coexistence of rhombohedral and tetragonal phases even at room temperature (see also Table 1). However, the tendency for multisplitting of this peak after the incorporation of the Fe/Mn into the NBT indicates an increased tetragonal phase content (see Table 1). The evolving shape of this peak as a result of E-poling points to the changing content of the rhombohedral/tetragonal phases (Table 1). Note that the shape of the diffraction peak near 2Θ≈38° evolved after application of the electric field. As its presence was evidence of rhombohedral (R3c) symmetry, this seemed to confirm this conclusion. In addition, the (110)_c_, (111)_c_ and (211)_c_ diffraction reflections (all split) of the rhombohedral symmetry evolved after E-poling, also indicating some changes in the rhombohedral structure. Indeed, the lattice constant (lattice volume) increased after the electric field application (Table 1).

This seems to be associated with the electric field transformation of the nonpolar/weak polar tetragonal regions to ferroelectric rhombohedral ones for the pure NBT [5,6] and in the opposite direction for the Fe- and Mn-doped NBT (except for 0.1%mol Fe-doped NBT, Table 1). The application of an electric field lead to an increased volume of the second phase (except for 0.5%mol Fe-doped NBT, Table 1).

The temperature evolution of the (110)_c_, (111)_c_, (200)_c_ and (211)_c_ Bragg reflections for the 0.5%mol Fe is shown in Figure 3.

The temperature variation in the patterns for the other compositions showed similar features. The split nature of the (110)_c_ and (111)_c_ reflections and the singlet nature the (200)_c_ reflection suggests rhombohedral symmetry. Closer inspection of Figure 3 shows that the (200)_c_ peak had a rather complex character, indicating that some tetragonal phase content (tetragonal regions) must have existed within the majority of the rhombohedral phase, even at room temperature. With increasing temperature, splitting of the (110)_c_ and (111)_c_ peaks gradually decreased, however it was still distinguishable until about 200 °C. Weak superlattice reflection occurred near 38° (indicated by an arrow), which had a Miller index 3/2 ½ ½ and is characteristic of the anti-phase rotation of octahedral TiO_6_ (corresponding to ā ā ā tilt system with space group R3c in Glazer notation [15,16]). This superlattice reflection persisted up to approximately 200 °C for unpoled samples and up to approximately 400–450 °C for poled samples. Distinct (200)_c_ peak splitting could be seen at about 350 °C and beyond, corresponding to tetragonal symmetry. Therefore, it can be inferred that the rhombohedral/tetragonal phases existed over a wide temperature interval up to 350 °C. In this temperature interval, the (111)_c_ peak was a mixture of double and single peaks. Our results also indicate that the cubic phase was not pure even at 600 °C.

The main effects of electric field application were a shift of the Bragg peaks to low angles and the support of the room-temperature profile of Bragg reflections to a higher temperature relative to the unpoled samples. This means that the lattice volume increased and the low-temperature ferroelectric state was protruded by E-poling.

Two anomalies of the temperature evolution of electric permittivity ε were visible (Figure 4), with a low-temperature anomaly (resembling a shoulder) visible at the so-called depolarization temperature T_d_ and a second one attributed to the maximum value of ε at T_m_. The frequency dispersion was more prominent in the vicinity of T_d_. The value of the electric permittivity increased after the Fe and Mn addition to the NBT (Figure 4).

The dielectric response can be divided into lattice deformation (intrinsic contribution, which is reversible) and displacement of the domain walls and/or the motion of the interphase boundaries (extrinsic contribution, which is usually irreversible). The improvement of the dielectric properties can be attributed to the effects of the Fe and Mn ions on the concentration of defects, lattice distortion, and the creation of microdefect complexes. In order to analyse the T_d_ phenomenon contribution to the electric permittivity, the extraction procedure proposed earlier [31] was applied. The fitting of the ε(T) function background was conducted outside the temperature region at which this bump appeared. The subtraction of the ε(T) and this fit gave the sought-after ∆ɛ_Td_(T) contribution to the electric permittivity (inserts in Figure 4). It was clearly seen from the inserts of Figure 4 that the Fe and Mn modification of the NBT and E-poling shifted T_d_ toward lower and higher temperatures, respectively (approximately 10 °C for both events) (Table 2). 

The position of the maximum of ε_Td_(T) increased with increasing frequency from approximately 10 °C/MHz for the pure NBT to approximately 35 °C/MHz for the Fe- and Mn-modified NBT. However, the position of the maximum decreased from approximately 10 °C/MHz to approximately 6 °C for the pure NBT and from 35 °C/MHz to approximately 20 °C/MHz for the Fe- and Mn-modified NBT as a result of E-poling. This means that the dielectric relaxation connected with the T_d_ phenomenon was enhanced by the NBT modification and deteriorated by the action of the electric field. This could be due to disturbance of the ferroelectric order by the modification and then its enhancement by the electric field.

Two anomalies/peaks of the DSC curves were visible (Figure 5). The first one was connected with the depolarization temperature T_d_ and related to the microtwinning process of the rhombohedral domains [7]. The second peak coincided with a rapid rise in the ε(T) curve, not with T_m_ (see Figure 4), and could be associated with the phase transformation from the coexisting rhombohedral and tetragonal phases [5,6] or from the orthorhombic phase to the tetragonal one [7]. Both peaks shifted to a higher temperature for the Fe- and Mn-modified NBT as well as after E-poling. In addition, the E-poling lead to sharper DSC(T) peaks. These results approximately coincided with the structural and dielectric measurement data.

The DC electric conductivity (σ_DC_) increased as a result of the Fe and Mn incorporation to the NBT (Figure 6).

Simultaneously, the activation energies deduced from Arrhenius plots decreased (Table 3).

These activation energies were markedly lower than the optical energy gap 3.2 eV [32] for the pure NBT and hence the electrical conductivity was due to impurities. Anomalies of the lnσ_DC_(1000/T) at approximately T_T-c_ (temperature of tetragonal-cubic phase transition), T_m_ and T_d_ (see Table 2) were clearly visible for all the investigated samples, which indicated changes in the band gap and/or changes in the scattering mechanism at different temperature intervals. The values 0.03–0.06 eV indicated that hopping charges [33,34] dominated at low temperatures. The values (0.24–0.29 eV), (0.61–0.65 eV) and (1.63–1.95 eV) suggested the possibility that conduction in the higher temperature interval for ionic charge carriers [33,34] may have been through the motion of the first and/or second ionization oxygen vacancies. The increase in conductivity in this temperature interval could be related to an increase in the concentration of the ionized vacancies. Note that the Fe- and Mn-doped NBT ceramics had two range sizes of grains: a larger one and a smaller one (Figure 1). Oxygen vacancies are easily accumulated at the grain of boundary as a path for conduction: a large grain will shorten the conduction path and conductivity increases, and vice versa. Competition between the influence of these two range sizes of grain on the ceramics’ behavior could partially lead to the increase in the electric conductivity.

Since manganese Mn is a multivalent element, it can be incorporated into NBT under different ionic valences (with coordination number VI): (1) as Mn^2+^ (r_Mn2+_ = 0.83 Å), (2) as Mn^3+^ (r_Mn3+_ = 0.65 Å) and (3) as Mn^4+^ (r_Mn4+_ = 0.53 Å) [35]. Considering the ionic radii of the different cations in the host lattice of NBT: Na^+^ (r_Na+_ = 1.39 Å with coordinator number XII), r_Bi3+_ = 1.17 Å with coordinator number VIII) and Ti^4+^ (r_Ti4+_ = 0.61 Å with coordinator number VI) [35], Mn is expected to replace the Ti cations. This doping creates extrinsic oxygen vacancies to compensate for the charge [36]: 2 Mn^2+^ = Mn″_Ti_ + V^••^_O_, 2 Mn^3+^ = 2 Mn′_Ti_ + V^••^_O_
(1)
where Mn″_Ti_ and Mn′_Ti_ represents Mn^2+^ and Mn^3+^, and V^••^_O_ denotes an oxygen vacancy. Substitution of Mn^4+^ in place of Ti^4+^ is charge neutral and leads to lattice contraction.

On the other hand, volatilization of A-site elements during sintering creates negatively charged A-site vacancies. To compensate for the charge equilibrium, intrinsic oxygen vacancies are created [36]: 2 Na* + O*_O_ = 2 V′_Na_ + V^••^_O_ + Na_2_O ↑, 2 Bi* + 3 O*_O_ = 2 V‴_Bi_ + 3 V^••^_O_ + Bi_2_O_3_ ↑(2)

It is expected that both intrinsic and extrinsic oxygen vacancies can act as charge carriers, which lead to increase dielectric conductivity, as seen in the present results.

Exposure of oxygen vacancies to an oxygen environment during cooling after the synthesizing process or annealing creates holes (h*) according to the relation [36]:1/2 O_2_ + V^••^_O_ = O*_O_ + 2 h* (3)

As the holes have higher mobility than oxygen vacancies, this leads to an additional increase in the electrical conductivity.

Substitution of Mn in A-sites seems a probable variant, since the ionic radius of Mn^2+^ ions is near to that of Na^+^ and Bi^3+^ and the ions have a small difference in valence. In this substitution, Mn can fill A-site vacancies created during A-site volatilization during sintering. This process decreases the density of the vacancies, which leads to a decrease in the electrical conductivity (in contrast to the present results).

Formation of microdefect dipoles such as (Mn′_Ti_-V^••^_O_) and (Mn″_Tii_-V^••^_O_) is possible [20], which can decrease the electric conductivity due to decreasing the density of oxygen vacancies and restricting their movement (in contrast to the present results).

It can be speculated that Mn^2+^ ions incorporated at the B-sites can absorb holes [8]. As a result, they can be oxidized to Mn^3+^ or Mn^4+^. This can occur according to the following relations [36]:Mn″_Ti_ + h* = Mn′_Ti_, Mn′_Ti_ + h*= Mn″_Ti_(4)

This substitution can reduce the electric conductivity by limiting the holes (in contrast to the present results).

Thus, the electrical conductivity data suggest rather that the Mn was incorporated in B-sites.

Iron mainly exists in Fe^2+^ (r_Fe2+_ = 0.78 Å), Fe^3+^ (r_Fe2+_ = 0.65 Å) and Fe^4+^(r_Fe4+_ = 0.59 Å) stable states (with coordinator number VI for all states) [35]. As the ionic radius of Fe^4+^ is near the ionic radius of Ti^4+^ (r_Ti4_+ = 0.61 Å) and both ions have the same ionic charge, Fe^4+^ can replace Ti^4+^, and this substitution is charge-neutral and does not sufficiently disturb the crystal lattice. Fe^3+^ added to NBT enters the B-sites due to the good correspondence in the ionic radii and the small difference in the ionic charge compared with Ti^4+^. This substitution will couple with an O^2−^ vacancy in the same unit cell and introduces excess negative charge, and the compensator should be positive charged. Oxygen vacancies (V_0_) can be considered as a compensator for excess charge (Fe^3+^ → Ti^4+^V^••^_O_). The vacancies can create defect pairs of Fe_Ti_ˈwith cation vacancies, which is typical for perovskites [20]. These defect pairs act as a donor. The crystal lattice volume would be decreased or increased because of Fe^4+^ or Fe^3+^ substitution for Ti^4+^, respectively. Additionally, the concentration of oxygen vacancies decreases as more Fe^3+^ is involved, resulting in an enlargement of the lattice volume. If one considers Fe^2+^ as impurity centers, substitution of Fe^2+^ → Na^+^ (r_Na+_ = 1.39 Å with coordinator number XII) or Bi^3+^ (r_Bi3+_ = 1.17 Å with coordinator number VIII) [35] is rather impossible. Here, there is a large difference in the ionic radii and a small difference in the valence. However, this substitution is postulated for perovskites [37,38,39,40]. This substitution introduces excess positive charge into the lattice, suppresses the formation of oxygen vacancies, induces the appearance of cation vacancies which compensate for charge imbalance and can lead to lattice contraction.

Taking into account the electronegativity of Fe (1.83) and Mn (1.55) and the electronegativity of the host ions Na (0.93), Bi (2.02) and Ti (1.54), one can conclude that both dopants should be incorporated in Ti-sites (the difference between the electronegativites of dopants and host ions is smallest for Ti-sites).

The room-temperature Raman spectra of the investigated samples are shown in Figure 7.

These spectra were assigned as belonging to the R3c phase. No contribution from a secondary phase was detected, most likely due to its small content. The spectra can be de-convolved into nine peaks using Lorentzian functions (insert of Figure 7a) and the frequencies and line widths of the modes are collected in Table 4. The frequencies were smaller than those obtained for the pure NBT [41].

The spectra of Fe- and Mn-modified NBT showed similar features, where broad bands were visible due to A-site disorder and overlapping Raman modes. In the range 100–1000 cm^−1^, there were four main regions connected with Na-O (~135 cm^−1^), Ti-O (~280 cm^−1^) and octahedral TiO_6_ (400–1000 cm^−1^) vibrations/rotations. The vibration frequency of Bi is below our experimental conditions. The rather complex character of the Ti-O band indicated that there was a coexistence of the rhombohedral and tetragonal phases in accordance with the X-ray results. There were variations in the shape and in the peak position of the Na-O and Ti-O bands as a result of the Fe/Mn addition to the NBT. Thus, one can conclude that Fe and Mn can occupy both A and B sites. There were also slight variations in the shape and position of the peak of the TiO_6_ band after the Fe/Mn modification of the NBT (see below). Using the relation ω = (k/µ)^1/2^, the Raman frequency ω was determined (k is force constant and µ is the reduced mass obtained from harmonic oscillator calculations). The ionic radii of Fe^3+^ and Fe^4+^ are larger and smaller, respectively, than that of Ti^4+^, which resulted in longer and shorter bond lengths. This in turn lead to smaller and higher force constants k, respectively. However, the relative atomic mass of Fe (55.84) is higher than that of Ti (47.87). Due to the larger ionic size of Fe^3+^ and higher atomic mass, the Ti-O band was expected to shift to lower frequency. However, the smaller ionic size of Fe^4+^ combined with its higher mass could shift the Ti-O band either in the lower or higher frequency direction, respectively. Shifting of this band to a lower frequency for the 0.5%mol Fe-doped NBT indicated the incorporation of Fe^3+^ and/or Fe^4+^ ions into B sites. However, this band seemed to shift to a higher frequency for the 1%mol Fe-doped NBT, indicating even more incorporation of Fe^4+^ ions into B sites and/or incorporation of Fe^2+^ ions into A sites. In the case of substitution of Fe^2+^ ions in A sites, their ionic radius is smaller than the Na ionic radius, and their relative mass is higher than that of Na, which can lead to the A-O band shifting to lower and higher frequencies, respectively. As this band seems to shift to a higher frequency, the influence of the differences in ionic radius is predominant. In general, the Raman scattering data confirmed the suggestions based on the XRD measurements. The octahedral TiO_6_ bands are associated with vibrations involving oxygen displacement and the existence of oxygen vacancies [6] and thus are independent of the mass of the cations. However, due to differences in the force constant caused by the larger or smaller ionic sizes of Fe^3+^ and Fe^4+^, respectively, some shifting of the TiO_6_ bands was expected. As more of these bands are shifted to lower frequency, the influence of Fe^4+^ ions is predominant. However, this conclusion is incomplete, because the TiO_6_ bands are also sensitive to the density of the oxygen vacancies, which changed after the incorporation of the Fe or Mn into the NBT. Although the reason for the change in the Raman spectra cannot be precisely assigned, there were detectable variations in the short-range crystalline order.

The Raman spectra of the poled samples were very similar to the unpoled ones (Figure 7), however, careful inspection showed some differences between them. The spectra for the poled samples were less broad and more symmetrical. This was due to the transformation of the tetragonal regions to predominantly rhombohedral ones for the pure NBT [5,6] and in the opposite direction for the Fe- and Mn-modified NBT, and to the ordering effect of the electric field. There was also a slight evolution in the shape and position of the TiO_6_ bands, which could be the result of the ordering influence of the electric field and redistribution of the oxygen vacancies.

Since the relation between the ionic radii and mass of multivalent Mn and the ionic radii and mass of Ti//Na is the same as for multivalent Fe, it was expected that variations in Na-O, Ti-O and TiO_6_ bands should be very similar. Indeed, careful analysis of the Raman data for the Mn-doped NBT confirmed this expectation.

The temperature evolution of the Raman spectra of the unpoled and poled 0.5%mol Fe is shown in Figure 8. The temperature-dependent spectra for other compositions exhibited similar features as are visible for that mentioned above. This is the reason why the temperature dependence of Raman spectra for the 1%mol Fe- and Mn-doped NBT were not presented here. Because of the ceramic form of the samples and the coexistence of the phases of different symmetry in the wide temperature interval, the obtained Raman modes were broadened and overlapped. Moreover, their temperature changes were characterized by the existence of slightly changeable broad bands in a very large temperature interval and some spectrum modifications near the phase transformations similar to that obtained for the pure NBT [42,43]. This feature is characteristic for relaxors and materials with a diffused phase transition [44]. Temperature-induced broadening of the main modes was visible, which could be the result of the evolution of the rhombohedral/tetragonal and tetragonal/cubic phases in the low- and high-temperature intervals, respectively. The Na-O and Ti-O bands’ positions shifted to a lower frequency with increasing temperature with a clearly visible change in the rate of this shift at temperatures close to T_d_ (the rate of shift was larger in the temperature interval from RT to T_d_ compared to the temperature interval above T_d_) (see also Table 2). There was also a considerable shift to a lower frequency of the high-frequency band (~530 cm^−1^), which was probably related to the expansion of the unit cell. This effect favors octahedral vibrations. At the same time, this band was gradually broadened. Simultaneously, the band at ~570 cm^−1^ was gradually broadened and shifted slightly to a higher frequency and seemed to disappear at about 280–300 °C, which may be connected with the gradual changes in the rhombohedral/tetragonal phase content. Note that the NBT05Fe ceramics possessed first-order Raman spectra even at 600 °C, where the average symmetry should be cubic. The Raman activity of the sample at high temperatures might be connected with the presence of weak-polar (tetragonal) nano- or meso-regions far above the tetragonal-cubic phase transformation. Some symptoms of this were observed in present XRD results.

The temperature evolution of the Raman spectra of the poled samples showed similar features as the unpoled ones (Figure 8). However, the main bands were more symmetric and less broad in the low-temperature range from RT to T_d_.

The temperature evolution of the Raman spectra of the Fe-/Mn-doped NBT indicated a rhombohedral-tetragonal and tetragonal-cubic sequence of phase transformations and exhibited temperatures T_d_ and 280 °C, characteristic of pure NBT, indicating a similar crystal structure and similar behavior with increasing temperature in accordance with the XRD results. This is further supported by temperature evolution of the mode wavenumbers, FWHM and integrated intensities (see below).

The temperature evolution of the mode wavenumbers, FWHM and integrated intensities of the unpoled and poled 0.5% mol Fe-doped NBT is shown in Figure 9, Figure 10 and Figure 11 (the same type of symbols is used for the particular mode for Figure 9, Figure 10 and Figure 11) (see also Table 4).

For both states, the wavenumbers of the modes were slightly temperature dependent. Only mode *a* showed a broad soft mode-like behavior shift up to about 150 °C (about 170 °C for poled state), and then very slightly shifted to a higher wavenumber with increasing temperature. This could be connected with the T_d_ phenomenon (see also Table 2 ). However, the modes *b*, *c* and *d* showed high wavenumber shift with temperature, almost a hard mode-like behavior, up to about 165 °C (up to about 180 °C for the poled state) related to the T_d_ phenomenon (see also Table 2). These modes, as well as the *e*, *f* and *g* modes, showed a deviation from a nearly linear behavior in the temperature interval 280–340 °C, pointing to a rhombohedral-tetragonal phase transition. This slight deviation could be related to the gradual increase in the ratio of the tetragonal phase volume fraction in this temperature range at the cost of the rhombohedral one. The wavenumbers of the majority of the modes show some kind of anomaly at about 520 °C, related to the tetragonal-cubic phase transformation. In contrast, both full widths at half-maximum FWHM (damping) and the integral intensity change significantly with increasing temperature. For the majority of the modes, after the initial growth, or nearly temperature independent of damping, significant step-like changes above about 130 °C were observed. Above about 280–290 °C, damping of the majority of the modes stabilized or slightly increased, which suggests that the sample became more symmetric at local scales with increasing temperature. In general, the integral intensity grew with increasing temperature up to T_d_, and then exhibited some kind of anomaly at the rhombohedral-tetragonal and tetragonal-cubic phase transformation temperatures. For poled samples, similar intensity features were observed which appeared more stable above the temperature of the rhombohedral-tetragonal phase transition.

Despite the increase in electric conductivity, the piezoelectric properties (d_33_ and k_33_ parameters) did not change after the incorporation of the Fe into the NBT. However, the Mn-doped NBT ceramics exhibited enhanced piezoelectric properties d_33_~89 and 97pC/N and k_33_~24 and 27% for 0.5 and 1% mol-doped NBT, respectively, compared with those of pure NBT (73 pC/N and 21%). This could be related to lattice distortion and perturbations of the local polarization and strain, which makes the reorientation of domains easier and leads to heterogeneous polar regions and additional interfacial energies [8,44].

## 4. Conclusions

Iron- and manganese-doped NBT ceramics with densities higher than 95% were prepared by the traditional solid-state reaction route. Their crystal structure, dielectric and thermal properties in the unpoled and poled state were examined. It was shown that the obtained materials possessed the same phase transition sequences, with the same characteristic temperatures, as pure NBT. The possibility of the Fe and Mn ions entering A and B sites was discussed in accordance with the radius/valence/electronegativity principles. The obtained results indicated that Fe and Mn could occupy both A and B sites. It was shown that Fe and Mn doping of NBT was an effective way to change their electric (electrical conductivity) and dielectric/piezoelectric properties. This included enhancing the electric permittivity, decreasing the depolarization temperature Td and increasing the piezoelectric coefficient d_33_, and increasing the electrical conductivity. In general, these modifications of the properties may be due to lattice disturbance, changes in the density of defects, formation of defect pairs, improvement of the domain reorientation conditions and instability of the local polarization state. It was proved that the crystal structure and dielectric response could be modulated by both Fe/Mn doping and by an applied electric field.

## Figures and Tables

**Figure 1 materials-15-06204-f001:**
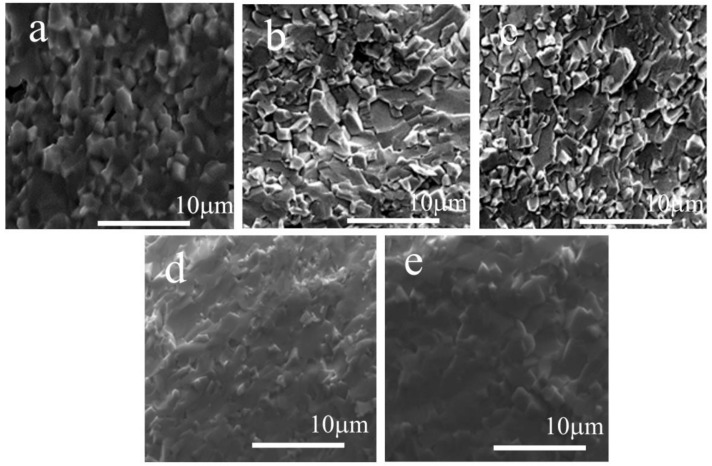
SEM micrographs of the surface of the investigated samples and SEM-EDS elemental mapping for selected samples: NBT (**a**), NBT05Mn (**b**), NBT1Mn (**c**), NBT05Fe (**d**) and NBT1Fe (**e**).

**Figure 2 materials-15-06204-f002:**
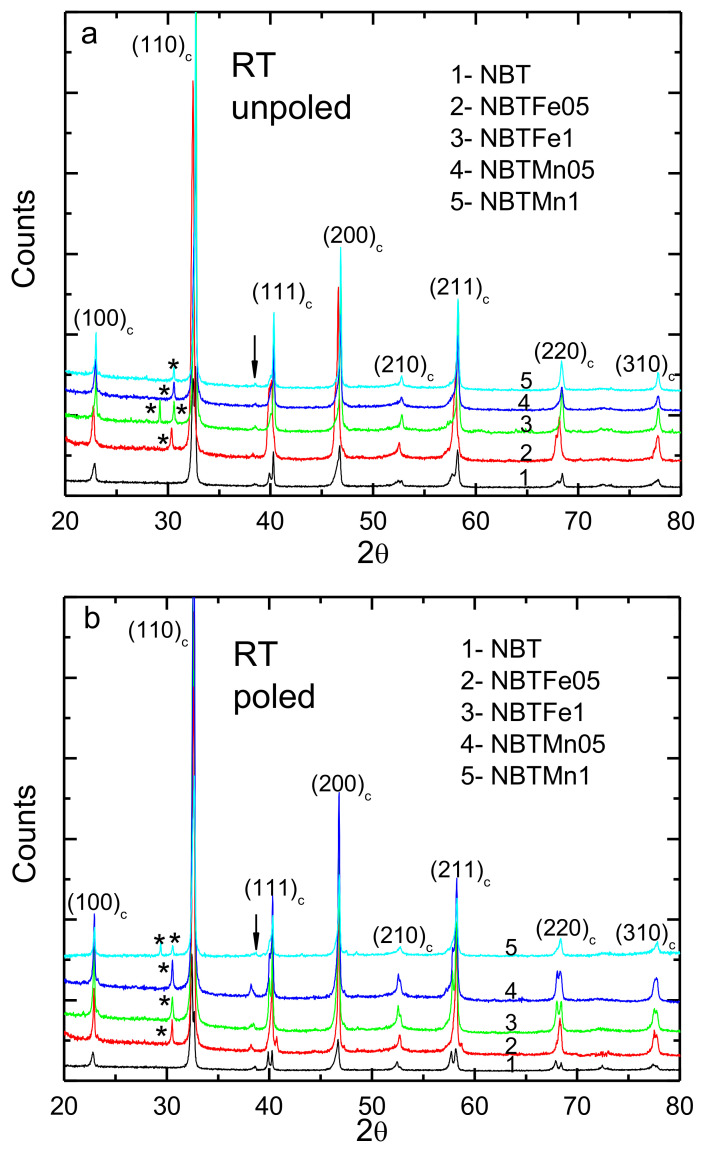
The room-temperature X-ray pattern of the investigated samples in (**a**) unpoled and (**b**) poled states. The “c” subscript corresponds to the cubic phase. The “*” shows secondary phase. The arrow shows the superlattice reflection of the rhombohedral R3c structure.

**Figure 3 materials-15-06204-f003:**
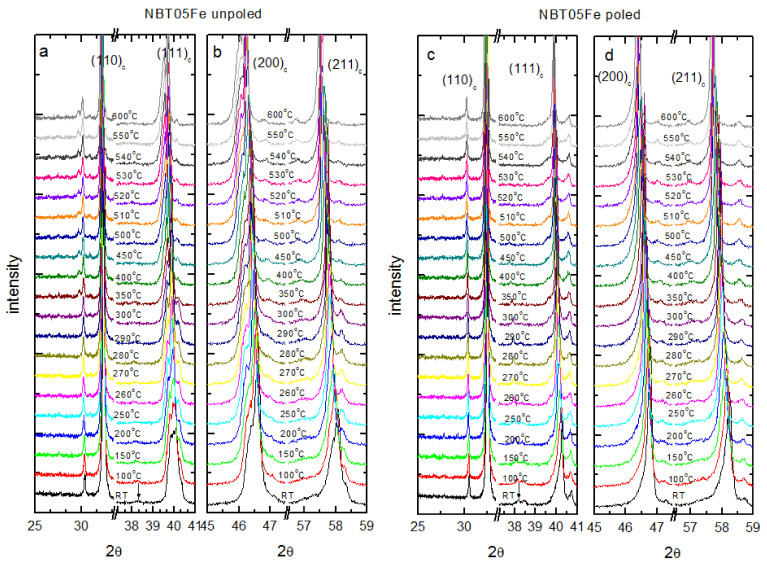
Temperature variation in the (110)_c_, (111)_c_ and (200)_c_ Bragg peaks of the NBT05Fe in (**a**,**b**) unpoled and (**c**,**d**) poled state. The “c” subscript corresponds to the cubic phase. The arrows show the superlattice reflection of the rhombohedral R3c structure.

**Figure 4 materials-15-06204-f004:**
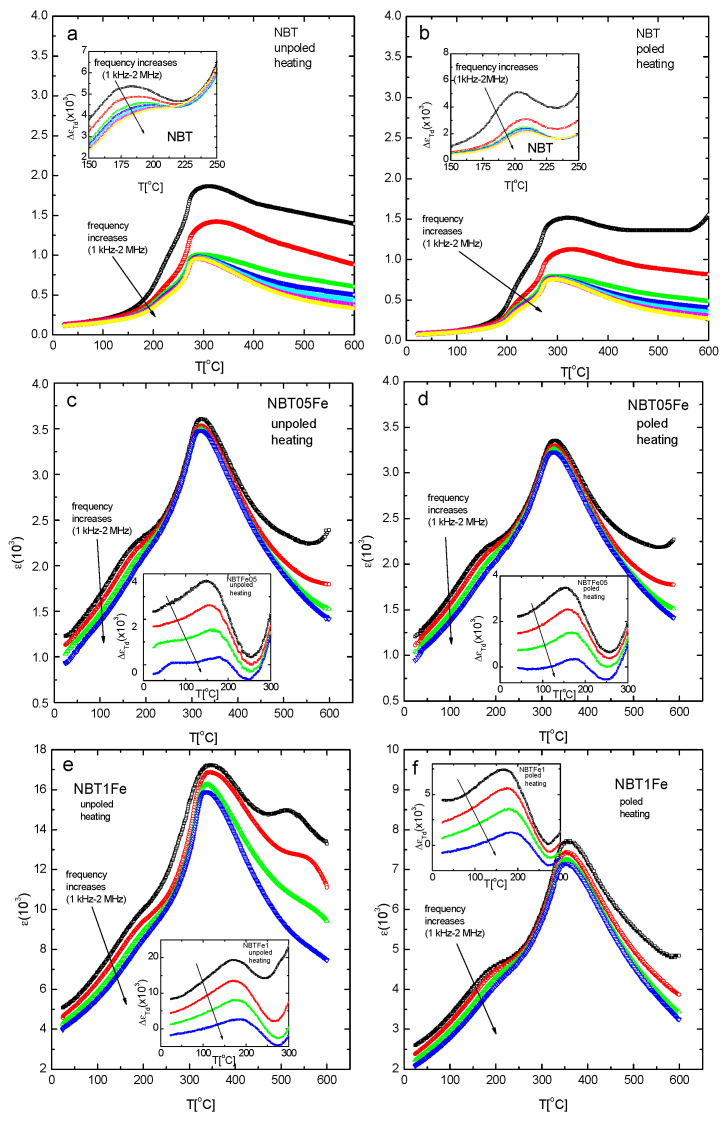
Temperature variation in electric permittivity of the investigated samples in unpoled and poled states. NBT unpoled (**a**) poled (**b**), NBT05Mn unpoled (**c**) poled (**d**), NBT1Mn unpoled (**e**) poled (**f**), NBT05Fe unpoled (**g**) poled (**h**) and NBT1Fe unpoled (**i**) poled (**j**).

**Figure 5 materials-15-06204-f005:**
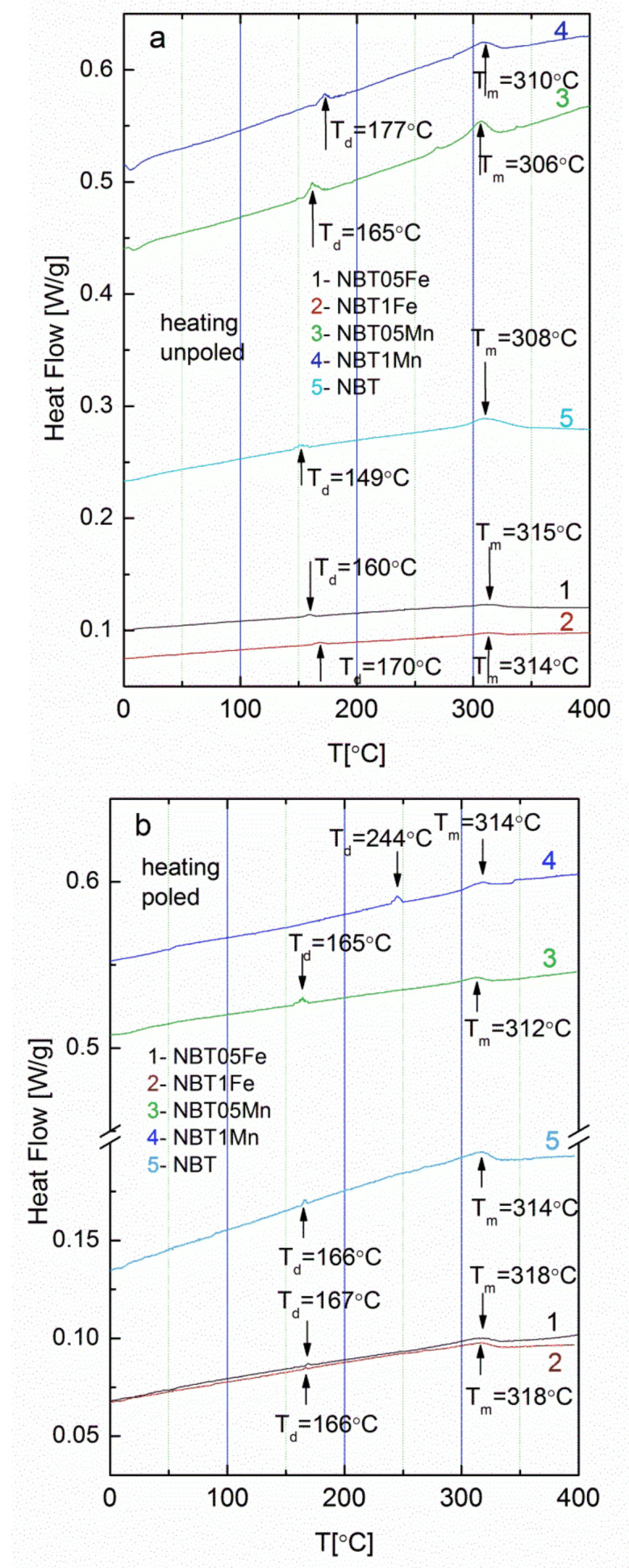
DSC curves of (**a**) unpoled and (**b**) poled samples.

**Figure 6 materials-15-06204-f006:**
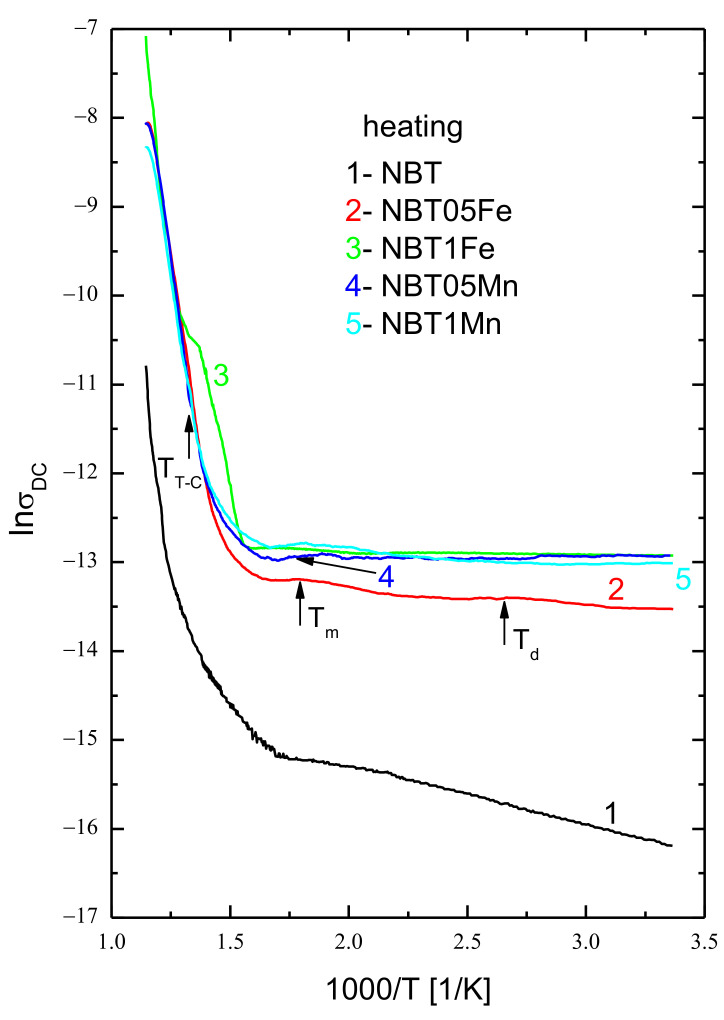
Evolution of DC electric conductivity with 1000/T of the investigated samples.

**Figure 7 materials-15-06204-f007:**
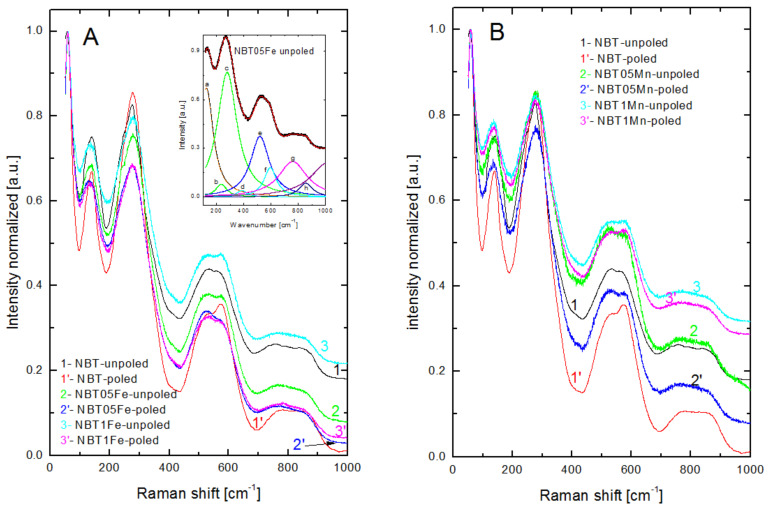
Room-temperature Raman spectra of the investigated samples in unpoled and poled states. Inset shows the spectral deconvolution of the room-temperature Raman spectrum of NBT05Fe. The a, b, c, d, e, f, g and h are mode names. (**A**) Fe-doped NBT, (**B**) Mn-doped NBT.

**Figure 8 materials-15-06204-f008:**
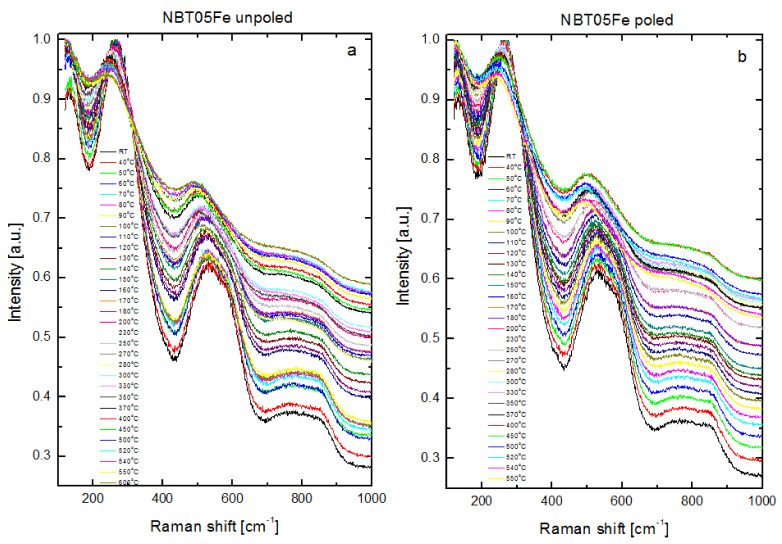
Temperature evolution of Raman spectra of NBT05Fe in (**a**) unpoled and (**b**) poled states.

**Figure 9 materials-15-06204-f009:**
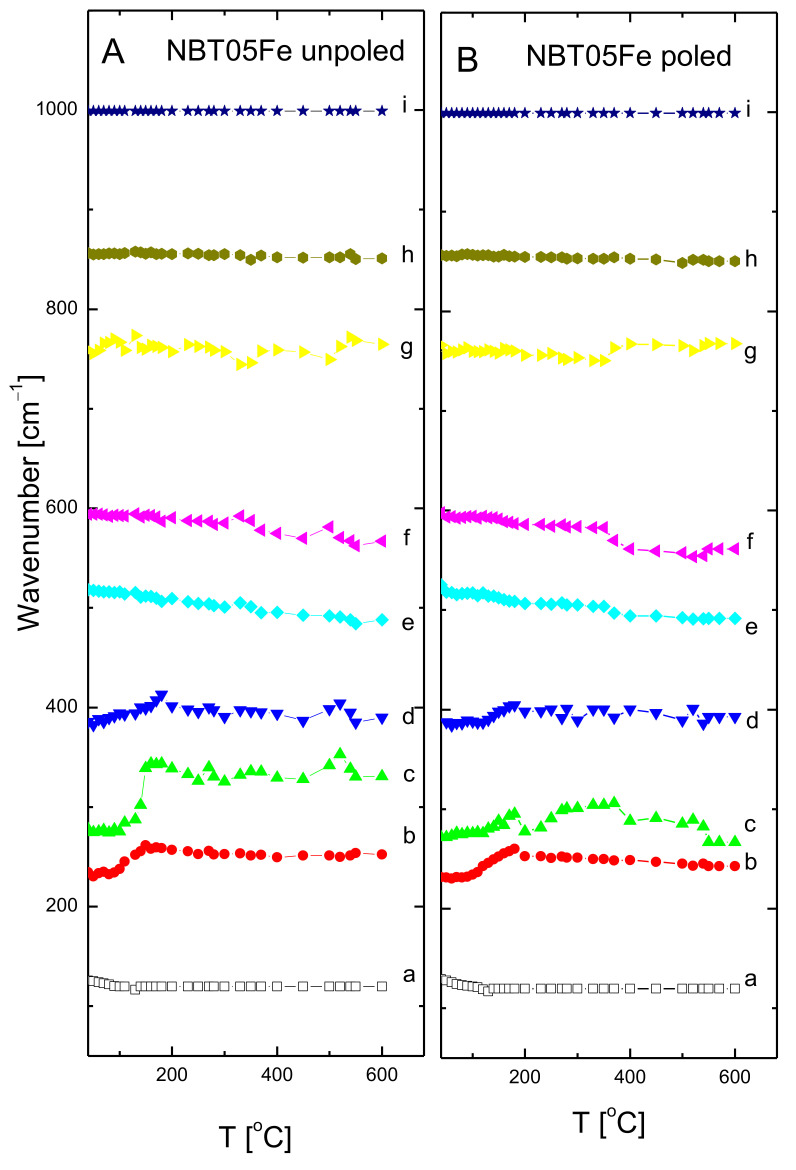
The temperature evolution of mode wavenumbers of unpoled and poled NBT05Fe. The same names for particular mode as in insert of Figure 7 are used. The a, b, c, d, e, f, g, h and i are mode names.

**Figure 10 materials-15-06204-f010:**
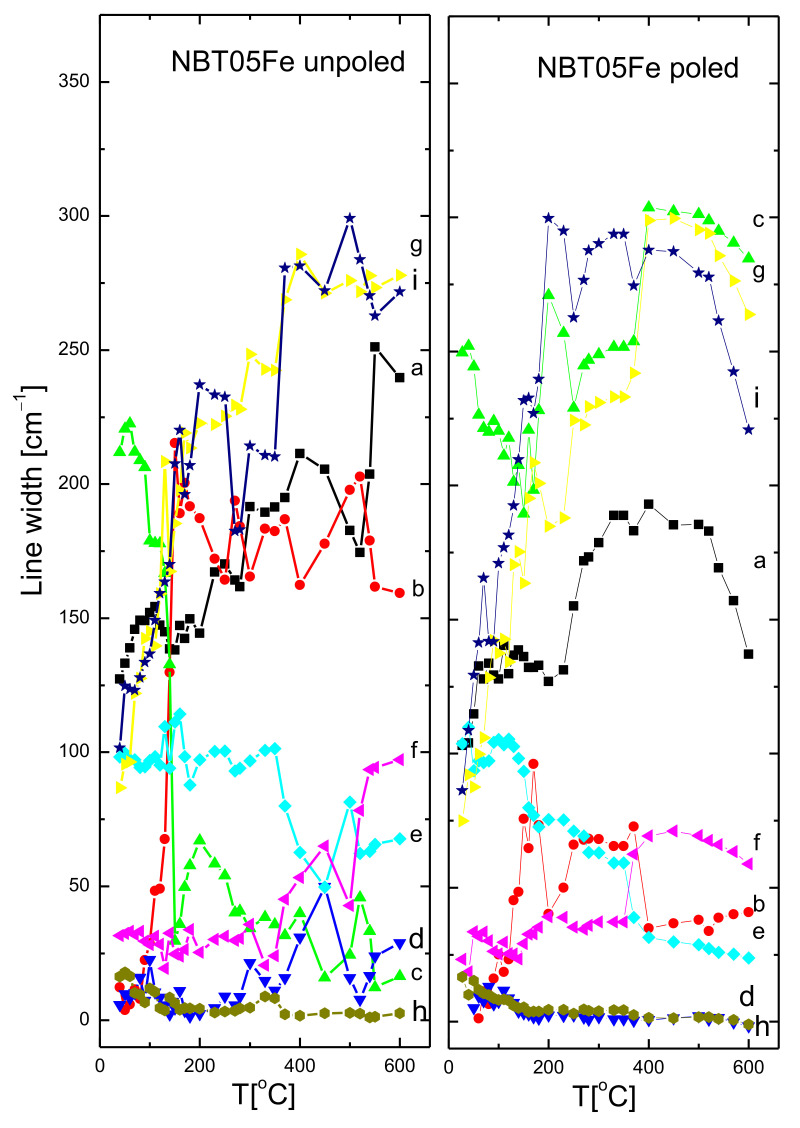
The temperature evolution of FWHM of unpoled and poled NBT05Fe. The a, b, c, d, e, f, g, h and i are mode names.

**Figure 11 materials-15-06204-f011:**
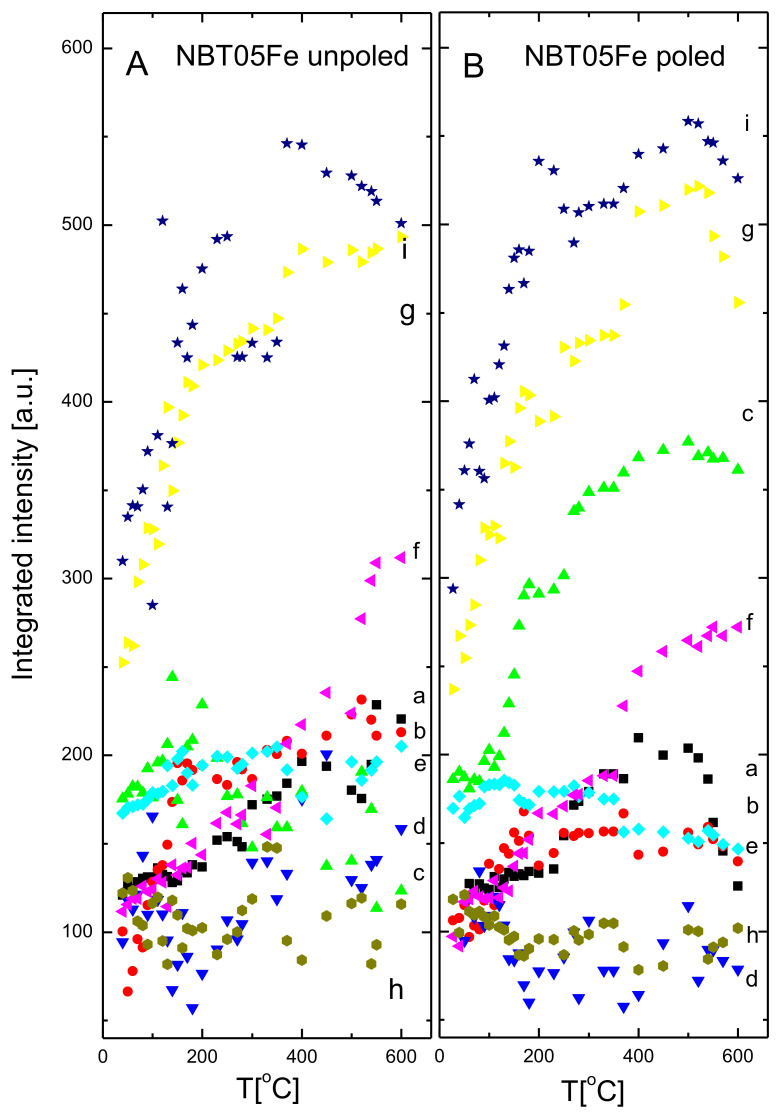
The temperature evolution of integrated intensities of unpoled and poled NBT05Fe. The a, b, c, d, e, f, g, h and i are mode names.

**Table 1 materials-15-06204-t001:** Rietveld refinement results of NBT and Fe- and Mn-modified NBT. V is lattice volume, and R is weighted profile.

Sample	Phase Composition	V/10^6^ pm^3^	R
NBT			
unpoled	R3C	353.07260	8.15%
poled	R3C	353.72130	7.93%
NBT05Fe			
unpoled	R3C 79.2%	351.98740	10.70%
	P4bm 16.1%	119.00890	10.91%
	Bi_5_Ti_3_FeO_15_ 4.7%	1202.07200	9.80%
poled	R3C 78.3%	351.72870	7.34%
	P4bm 18.4%	119.24740	7.74%
	Bi_5_Ti_3_FeO_15_ 3.3%	1208.56100	7.11%
NBT1Fe			
unpoled	R3C 74.2%	352.14380	7.34%
	P4bm 25.0%	120.00790	7.24%
	Bi_5_Ti_3_FeO_15_ 0.8%	1217.23200	8.86%
poled	R3C 81.3%	352.87970	3.79%
	P4bm 13.9%	120.12210	5.04%
	Bi_5_Ti_3_FeO_15_ 4.8%	1210.36900	6.51%
NBT05Mn			
unpoled	R3C 47.1%	351.92630	3.98%
	P4bm 52.0%	119.33060	3.32%
	Bi_5_Ti_3_FeO_15_ 0.9%	1204.94400	7.39%
poled	R3C 46.6%	352.17970	6.32%
	P4bm 52.3%	118.94330	7.87%
	Bi_5_Ti_3_FeO_15_ 1.1%	1205.90500	9.98%
NBT1Mn			
unpoled	R3C 69.4%	351.74620	7.33%
	P4bm 30.1%	119.00110	7.80%
poled	Bi_5_Ti_3_FeO_15_ 0.5%	1203.90000	10.47%
	R3C 45.6%	351.91730	4.01%
	P4bm 53.4%	118.99880	4.90%
	Bi_5_Ti_3_FeO_15_ 1.0%	1212.02800	6.72%

**Table 2 materials-15-06204-t002:** T_d_ and T_m_ of investigated samples.

Sample	T_d_ [°C]	T_m_ [°C]
NBT		
unpoled	190	316
poled	196	317
NBT05Fe		
unpoled	160	318
poled	168	322
NBT1Fe		
unpoled	165	338
poled	170	351
NBT05Mn		
unpoled	165	290
poled	195	293
NBT1Mn		
unpoled	177	306
poled	244	312

**Table 3 materials-15-06204-t003:** Activation energy of investigated materials estimated from DC electric conductivity measurements.

Temperature Range	RT-190 °C	190–320 °C	320–450 °C	450–540 °C	540–600 °C
	**Activation Energy**				
	**E_a_[eV]**	**E_a_[eV]**	**E_a_[eV]**	**E_a_[eV]**	**E_a_[eV]**
NBT	0.057	0.049	0.290	0.690	1.951
NBT05Fe	0.041	0.031	0.240	0.630	1.765
NBT1Fe	0.039	0.041	0.280	0.670	1.692
NBT05Mn	0.045	0.036	0.270	0.630	1.633
NBT1Mn	0.039	0.027	0.261	0.610	1.722

**Table 4 materials-15-06204-t004:** Notations, frequencies and line widths of Raman spectra of NBT05Fe ceramics at room temperature.

Mode Name	Frequency (cm^−1^)	FWHM (cm^−1^)
a	126	120
b	234	100
c	278	176
d	386	94
e	518	167
f	594	112
g	759	253
h	856	122
i	999	310

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
