# Peer review of "Properties of Na0.5Bi0.5TiO3 Ceramics Modified with Fe and Mn"

_materials, 2022, doi:10.3390/ma15186204_

Round 1
Reviewer 1 Report
This is a technical well done work. However, the authors were not able to demonstrate sufficient novelty of their results. Both Fe [Ref. 15, Y. Wan et al. Material Science (Poland) 2009;27(2):471-476, https://doi.org/10.1016/j.ceramint.2014.10.066, https://doi.org/10.1111/jace.16401, https://doi.org/10.1007/s10948-019-05163-z] and Mn doping [Ref. 15, https://doi.org/10.1111/j.1551-2916.2009.03551.x, https://doi.org/10.1016/j.matlet.2016.09.105] of NBT are well known. The enhancement of piezoelectric properties by Fe doping was already reported in [Ref. 15], by Mn in [https://doi.org/10.1111/j.1551-2916.2009.03551.x]. A wide region of rhombohedral - tetragonal phase coexistence was reported earlier in [https://doi.org/10.1063/1.3645054]. The temperature variation of dielectric permittivity for undoped and Fe-doped (0.5 and 1 mol%) was reported by the same authors in [https://doi.org/10.1080/01411594.2020.1804902]. Here, also DSC curves were given. The decrease of the phase transition temperature with Fe doping was obtained by the authors in the same study, the similar effect by Mn-doping in [https://doi.org/10.1111/j.1551-2916.2009.03551.x]. The description of FeTi-VO and MnTi-VO defects is not new. Therefore, it should be provided with references. Moreover, earlier work of Raman scattering was inadequately acknowledged. Thus, this paper should be rejected in the present form.
The main shortages of the manuscript are:
1) The introduction suffers from the absence of an analysis of the state-of-the-art of understanding of NBT properties (cf., e.g., https://doi.org/10.1063/1.3115409, https://doi.org/10.1063/1.3645054, https://doi.org/10.1063/1.4950790, etc.). Also, the authors do not acknowledge their own previous work, e.g. on Fe-doping (0, 0.5 and mol 1 %)of NBT [https://doi.org/10.1080/01411594.2020.1804902] yielding similar results of dielectric permittivity and DSC peaks.
2) With regard of the current understanding of dielectric properties of NBT-based materials, three dielectric anomaly peaks at Td, Ts and Tm should be considered. Here, Td is the depolarization temperature, i.e. the temperature of the steepest decrease of remanent polarization, denoting a ferroelectric-relaxor transition since above Td, the ferroelectric long range order is no longer maintained. Ts denotes a “shoulder”-like low temperature dielectric anomaly above which the polar nanoregions are subjected to large fluctuations making the formation of a ferroelectric phase difficult, even under high electric fields. At temperature Tm the dielectric permittivity reaches the maximum value. Tm is attributed to a diffuse phase transition of lower symmetry polar nanoregions (R3c) into higher symmetry (P4bm) ones, and a their thermal evolution. Correspondingly Td and Tm should be seen in DSC.
3) The experimental data should be explained with a comprehensive model. For instance, dielectric properties should be related to DSC measurements, and, thus, to the corresponding phase transitions at Td and Tm. Here, a table of Tm values would be useful for the reader.
4) The attribution of conductivity activation energies to differents mechanisms should be provided with references. Also, the defect reactions, equations (1) to (4).
5) The Raman frequencies of Na-O and Ti-O in table 3 were assigned to the corresponding vibrations without any references to previous work. For the TiO6 octahedra the reference was given only in the following text.
6) Some results of minor importance for the under standing of the paper (Figures 3, 4, 9, 10-12) could be moved to Supplemental Material.
Author Response
Referee 1. Comments and Suggestions for Authors
This is a technical well done work. However, the authors were not able to demonstrate sufficient novelty of their results. Both Fe [Ref. 15, Y. Wan et al. Material Science (Poland) 2009;27(2):471-476, https://doi.org/10.1016/j.ceramint.2014.10.066, https://doi.org/10.1111/jace.16401, https://doi.org/10.1007/s10948-019-05163-z] and Mn doping [Ref. 15, https://doi.org/10.1111/j.1551-2916.2009.03551.x, https://doi.org/10.1016/j.matlet.2016.09.105] of NBT are well known. The enhancement of piezoelectric properties by Fe doping was already reported in [Ref. 15], by Mn in [https://doi.org/10.1111/j.1551-2916.2009.03551.x]. A wide region of rhombohedral - tetragonal phase coexistence was reported earlier in [https://doi.org/10.1063/1.3645054]. The temperature variation of dielectric permittivity for undoped and Fe-doped (0.5 and 1 mol%) was reported by the same authors in [https://doi.org/10.1080/01411594.2020.1804902]. Here, also DSC curves were given. The decrease of the phase transition temperature with Fe doping was obtained by the authors in the same study, the similar effect by Mn-doping in [https://doi.org/10.1111/j.1551-2916.2009.03551.x]. The description of FeTi-VO and MnTi-VO defects is not new. Therefore, it should be provided with references. Moreover, earlier work of Raman scattering was inadequately acknowledged. Thus, this paper should be rejected in the present form.
Reply:
- Only a few papers connected Fe and Mn-doped NBT exist in literaturę (mentioned by Referee, included in improved version of present paper ). These papers contained mainly some properties of unpoled these materials at room temperature. It is well known that measurements of poled samples are essential. So, new measurements in unpoled/poled state in wide temperaturę range are necessary.
- Some results (mainly dielectric and DSC) connected with undoped and Fe/Mn- doped NBT was reported by us in short two papers in Phase Transitions (included in improved version of present paper). These papers included mainly some results of measurements at room temperature with their short description and discussion. The present paper contains more types of measurements in wide temperaturę range with expanded description, expanded discussion and conclusion.
- The description of FeTi-VOand MnTi-VO defects was provided with references in improved version of present paper.
The main shortages of the manuscript are:
- The introduction suffers from the absence of an analysis of the state-of-the-art of understanding of NBT properties (cf., e.g., https://doi.org/10.1063/1.3115409, https://doi.org/10.1063/1.3645054, https://doi.org/10.1063/1.4950790, etc.). Also, the authors do not acknowledge their own previous work, e.g. on Fe-doping (0, 0.5 and mol 1 %)of NBT [https://doi.org/10.1080/01411594.2020.1804902] yielding similar results of dielectric permittivity and DSC peaks.
Revised as suggested.
- With regard of the current understanding of dielectric properties of NBT-based materials, three dielectric anomaly peaks at Td, Tsand Tm should be considered. Here, Td is the depolarization temperature, i.e. the temperature of the steepest decrease of remanent polarization, denoting a ferroelectric-relaxor transition since above Td, the ferroelectric long range order is no longer maintained. Ts denotes a “shoulder”-like low temperature dielectric anomaly above which the polar nanoregions are subjected to large fluctuations making the formation of a ferroelectric phase difficult, even under high electric fields. At temperature Tm the dielectric permittivity reaches the maximum value. Tm is attributed to a diffuse phase transition of lower symmetry polar nanoregions (R3c) into higher symmetry (P4bm) ones, and a their thermal evolution. Correspondingly Td and Tm should be seen in DSC.
Revised as suggested.
- The experimental data should be explained with a comprehensive model. For instance, dielectric properties should be related to DSC measurements, and, thus, to the corresponding phase transitions at Tdand Tm. Here, a table of Tm values would be useful for the reader.
Revised as suggested.
- The attribution of conductivity activation energies to differents mechanisms should be provided with references. Also, the defect reactions, equations (1) to (4).
Revised as suggested.
- The Raman frequencies of Na-O and Ti-O in table 3 were assigned to the corresponding vibrations without any references to previous work. For the TiO6octahedra the reference was given only in the following text.
Revised as suggested.
- Some results of minor importance for the under standing of the paper (Figures 3, 4, 9, 10-12) could be moved to Supplemental Material.
Reply: In our opinion, the work is more transparent and friendly for reader in present form (reaching out to Supplement is troublesome and distractful).
Reviewer 2 Report
The influence of Fe and Mn dopants on the structure and properties of one of the most actively studied lead-free ferroelectrics, sodium-bismuth titanate, was studied in the manuscript. The crystal structure, dielectric and thermal properties of these ceramics were measured in both unpoled and poled states. It can be clearly seen that the Fe/Mn doping and E-poling offers an effective way to modify the NBT properties. However, there are a number of questions and comments:
1. The literature review extension, adding references and recent information about BNT studies (for example, DOI: 10.1080/02603594.2020.1813728, DOI: 10.1134/S0020168520010136), including those published in MDPI journals (for example, doi: 10.3390/cryst90402 and many others) is recommended.
2. Fig. 1 a-e - there are no explanations either in the caption to the figure or in the text of the manuscript, what each of the figures refers to. Besides, Fig. 1d is of poor quality.
3. There is a doubtful explanation for the change in parameters: «The change of lattice parameters can be mainly due to the difference in ionic size of the dopant substituent in comparison to the host ion, and their multiple oxidation states, which leads to the different bond length of Fe-O compared to the Ti-O and Na/Bi-O bonds, and changed force constants. The initial increase of lattice volume (V) could be associated with predominant substitution of Ti4+ by the larger Fe3+, and the later decrease of V with the alternative substitution of Ti4+ by smaller Fe4+ and eventually (Na/Bi)2+ by smaller Fe2+ (see below)». - If the transition of iron to +4 in the presence of alkali cations can still be assumed, then the substitution of (Na/Bi)2+ by Fe2+ is improbable due to the colossal differences in ionic radii (Fe2+ 0.92, Na+ ~Bi3+ =1.51 Å). There are no oxygen compounds where Fe2+ exhibits cuboctahedral coordination, for example, in ilmenite FeTiO3 it has a coordination number of 6, so the replacement of (Na/Bi)2+ by smaller Fe2+ without destroying the perovskite structure is impossible.
4. «The complex character of the (200)c peak for pure NBT indicates the coexistence of rhombohedral and tetragonal phases even at room temperature (see also Table 1).» - why the tetragonal phase exactly, and not orthorhombic, monoclinic or some other, for example?
5. Fig. 4 duplicates the data presented in fig. 3, which are already well interpreted; there is no necessity to include it in the text.
6. Ionic radii are given (“…Mn3+ (rMn3+=0.65Å ) and (3) as Mn4+ (rMn4+=0.53Å ). Considering the ionic radii of different cations in the host lattices of NBT: Na+ (rNa+=1.02Å ), rBi3+=1.03Å ), and Ti4+ (rTi4+=0.61Å ), Mn is expected to replace the Ti cations….”), but firstly, they do not indicate which coordination number they belong to, and secondly, Shannon's table of ionic radii is now commonly used http://abulafia.mt.ic.ac.uk/shannon/ptable.php
7. «It is possible that Mn could occupy the A-site vacancies, but…» - this is impossible, based on ionic radii and coordination preferences.
8. «If one considers Fe2+ as impurity centers, substitution of Fe2+→Na+ (rNa+=1.02Å ) or Bi3+ (rBi3+=1.03Å ) seems to be possible.» - this is extremely doubtful, based on ionic radii and coordination preferences.
Author Response
Referee 2. Comments and Suggestions for Authors
The influence of Fe and Mn dopants on the structure and properties of one of the most actively studied lead-free ferroelectrics, sodium-bismuth titanate, was studied in the manuscript. The crystal structure, dielectric and thermal properties of these ceramics were measured in both unpoled and poled states. It can be clearly seen that the Fe/Mn doping and E-poling offers an effective way to modify the NBT properties. However, there are a number of questions and comments:
- The literature review extension, adding references and recent information about BNT studies (for example, DOI: 10.1080/02603594.2020.1813728, DOI: 10.1134/S0020168520010136), including those published in MDPI journals (for example, doi: 10.3390/cryst90402 and many others) is recommended.
Revised as suggested.
- 1 a-e - there are no explanations either in the caption to the figure or in the text of the manuscript, what each of the figures refers to. Besides, Fig. 1d is of poor quality.
Reply: The explanations of Fig. 1 a-e were done in the captions. Quality of Fig. 1d was improved.
- There is a doubtful explanation for the change in parameters: «The change of lattice parameters can be mainly due to the difference in ionic size of the dopant substituent in comparison to the host ion, and their multiple oxidation states, which leads to the different bond length of Fe-O compared to the Ti-O and Na/Bi-O bonds, and changed force constants. The initial increase of lattice volume (V) could be associated with predominant substitution of Ti4+ by the larger Fe3+, and the later decrease of V with the alternative substitution of Ti4+ by smaller Fe4+ and eventually (Na/Bi)2+ by smaller Fe2+ (see below)». - If the transition of iron to +4 in the presence of alkali cations can still be assumed, then the substitution of (Na/Bi)2+ by Fe2+ is improbable due to the colossal differences in ionic radii (Fe2+ 0.92, Na+ ~Bi3+ =1.51 Å). There are no oxygen compounds where Fe2+ exhibits cuboctahedral coordination, for example, in ilmenite FeTiO3 it has a coordination number of 6, so the replacement of (Na/Bi)2+ by smaller Fe2+ without destroying the perovskite structure is impossible.
Reply: We agree with the Referee: the second sentence was removed.
- «The complex character of the (200)c peak for pure NBT indicates the coexistence of rhombohedral and tetragonal phases even at room temperature (see also Table 1).» - why the tetragonal phase exactly, and not orthorhombic, monoclinic or some other, for example?
Reply: According to many experimental results, rhombohedral and tetragonal phases coexist in wide temperaturę range. On the other hand, TEM measurements showed that rhombohedral-tetragonal phase transition may go through modulated orthorhombic phase [Chem.Mater. 2008;20:5061]. However, it should be noted that the existence of the orhorhombic phase has not been fully confirmed. [Appl.Phys.Lett. 2018;113:032901]. In addition, manner in which samples are prepared for the TEM measurements is destructive and can cause a change of crystal structure [J.Alloys and Comp. 2022;911:165104]. According to our knowledge, any other symmetry existence in NBT was not detected.
- 4 duplicates the data presented in Fig. 3, which are already well interpreted; there is no necessity to include it in the text.
Reply: Fig. 3 and adeuquate text was removed.
- Ionic radii are given (“…Mn3+ (rMn3+=0.65Å ) and (3) as Mn4+ (rMn4+=0.53Å ). Considering the ionic radii of different cations in the host lattices of NBT: Na+ (rNa+=1.02Å ), rBi3+=1.03Å ), and Ti4+ (rTi4+=0.61Å ), Mn is expected to replace the Ti cations….”), but firstly, they do not indicate which coordination number they belong to, and secondly, Shannon's table of ionic radii is now commonly used http://abulafia.mt.ic.ac.uk/shannon/ptable.php
Reply: Improved as suggested.
- «It is possible that Mn could occupy the A-site vacancies, but…» - this is impossible, based on ionic radii and coordination preferences.
Reply: We agree with the Referee- the sentence was removed.
- «If one considers Fe2+ as impurity centers, substitution of Fe2+→Na+ (rNa+=1.02Å ) or Bi3+ (rBi3+=1.03Å ) seems to be possible.» - this is extremely doubtful, based on ionic radii and coordination preferences.
Reply: In general, we agree with Referee, however, possibility incorporation of Fe ion into B and A-sites of perovskites is commonly postulated. This is mainly because in many cases interpretation of experimental results is impossible without it. Finaly, 1. we changed this sentence: „If one considers Fe2+ as impurity centers, substitution of Fe2+→Na+ (rNa+=1.39Å, with coordinator number XII) or Bi3+ (rBi3+=1.17Å, with coordinator number VIII) is rather impossible”., 2. we added the sentence: “However, this substitution is postulated for perovskites.”.
Reviewer 3 Report
The authors studied the crystal structure, dielectric and thermal properties of Na0.5Bi0.5TiO3 and Fe and Mn-modified Na0.5Bi0.5TiO3 (0.5 and 1 mol%) ceramics. The paper is written well and the experimental data are rich and complete. This paper can be accepted after minor revision.
The EDS (or EDX) should be provide in this work.
Discussions of relevant literature on structure, SEM and EDS could be further enhanced, which can link to the existing work. Authors might consider the following relevant recent work in this regard: https://doi.org/10.1063/5.0078188.
Some SEM pictures are very blurred (e.g. a and e), please take relevant pictures again.
Author Response
Refetree 3. Comments and Suggestions for Authors
The authors studied the crystal structure, dielectric and thermal properties of Na0.5Bi0.5TiO3 and Fe and Mn-modified Na0.5Bi0.5TiO3 (0.5 and 1 mol%) ceramics. The paper is written well and the experimental data are rich and complete. This paper can be accepted after minor revision.
- The EDS (or EDX) should be provide in this work.
Improved as suggested.
- Discussions of relevant literature on structure, SEM and EDS could be further enhanced, which can link to the existing work. Authors might consider the following relevant recent work in this regard: https://doi.org/10.1063/5.0078188.
Impropved asd suggested.
- Some SEM pictures are very blurred (e.g. a and e), please take relevant pictures again.
Improved as suggested.
Round 2
Reviewer 1 Report
The authors have satisfactorly taken into account my comments. However, there are still some minor correction to be made:
- replace rows 58-66 introducing Td and Tm before 47-48
- formate value and dimension for all values as required by the journal, sometime they are separated by space, sometime not
- row 265: DC
- row 303: space before [35]
- rows 315-318, 353-354: give a reference to ion radii since different sources give different values
row 545: formate d33
row 629: formate doi
Author Response
Changes have been made in accordance with the reviewer's recommendations:
- replace rows 58-66 introducing Td and Tm before 47-48
Replay: improved as suggested
- formate value and dimension for all values as required by the journal, sometime they are separated by space, sometime not
Replay: improved as suggested
- row 265: DC
Repleay: improved as suggested
- row 303: space before [35]
Repleay: improved as suggested
- rows 315-318, 353-354: give a reference to ion radii since different sources give different values
Replay: improved as suggested
- row 545: formate d33
Replay: improved as suggested
- row 629: formate doi
Replay: improved as suggested
Best regards
Jan Suchanicz
Reviewer 2 Report
I think that this article, after the corrections already made by the authors, can be published, but the answer regarding Fe2+ in positions A "...However, this replacement is postulated for perovskites..." raises doubts, please provide links to such articles.
Author Response
I think that this article, after the corrections already made by the authors, can be published, but the answer regarding Fe2+ in positions A "...However, this replacement is postulated for perovskites..." raises doubts, please provide links to such articles.
Replay: improved as suggested. An additional citations: [39] [40] [41] [42] concerning Fe2+ in A positions in the perovskites has been introduced in the manuscript.